# Instantaneous Pre-Fire Biomass and Fuel Load Measurements from Multi-Spectral UAS Mapping in Southern African Savannas



Tom Eames [1,*], Jeremy Russell-Smith [2], Cameron Yates [2], Andrew Edwards [2], Roland Vernooij [1], Natasha Ribeiro [3], Franziska Steinbruch [4] and Guido R. van der Werf [1]

1   Department of Earth Sciences, Faculty of Science, Vrije Universiteit Amsterdam, 1081 HV Amsterdam, The Netherlands; r.vernooij@vu.nl (R.V.); g.r.vander.werf@vu.nl (G.R.v.d.W.)
2   Charles Darwin University, P.O. Box 40146, Casuarina, Darwin NT 0811, Australia; Jeremy.Russell-Smith@cdu.edu.au (J.R.-S.); Cameron.Yates@cdu.edu.au (C.Y.); Andrew.Edwards@cdu.edu.au (A.E.)
3   Department of Forest Engineering, Faculty of Agronomy and Forest Engineering, Universidade Eduardo Mondlane, Avenida Julius Nyerere, Street. nr. 3453, Maputo, Mozambique; joluci2000@yahoo.com
4   Wildlife Conservation Society Mozambique, Orlando Mendes Street, no.163, Sommerschield, Maputo, Mozambique; franziska.steinbruch@gmail.com
*   Correspondence: t.c.eames@vu.nl

**Abstract:** Landscape fires are substantial sources of (greenhouse) gases and aerosols. Fires in savanna landscapes represent more than half of global fire carbon emissions. Quantifying emissions from fires relies on accurate burned area, fuel load and burning efficiency data. Of these, fuel load remains the source of the largest uncertainty. In this study, we used high spatial resolution images from an Unmanned Aircraft System (UAS) mounted multispectral camera, in combination with meteorological data from the ERA-5 land dataset, to model instantaneous pre-fire above-ground biomass. We constrained our model with ground measurements taken in two locations in savanna-dominated regions in Southern Africa, one low-rainfall region (660 mm year$^{-1}$) in the North-West District (Ngamiland), Botswana, and one high-rainfall region (940 mm year$^{-1}$) in Niassa Province (northern Mozambique). We found that for fine surface fuel classes (live grass and dead plant litter), the model was able to reproduce measured Above-Ground Biomass (AGB) ($R^2$ of 0.91 and 0.77 for live grass and total fine fuel, respectively) across both low and high rainfall areas. The model was less successful in representing other classes, e.g., woody debris, but in the regions considered, these are less relevant to biomass burning and make smaller contributions to total AGB.

**Keywords:** burning; biomass burning; fuel load; savanna fire; drone; UAS; remote sensing

## 1. Introduction

Landscape fires contribute to the atmospheric budgets of aerosols and elevate greenhouse gas levels. While all fires emit $CO_2$, fires burning in deforestation zones and tropical peatlands lead to net $CO_2$ emissions. Fires also emit methane and nitrous oxide, with respective global warming potentials of 28 and 265 times greater than $CO_2$ over 100 years [1]. The African continent plays host to a large number of these fires, and the portion of the continent in the Southern Hemisphere alone contributes almost a third of global emissions attributable to wildfires [2] . Of these, most (>90%) come from tropical savannas, a biome where fire disturbance is recognised as affecting ecosystem structure, function and dynamics [3,4]. Savanna fires consume fuels predominantly at the ground level [5], comprising mostly grasses, leaf litter and other pieces of woody debris or dead vegetation deposited on the surface. Trees and larger shrubs are rarely affected unless the fire is intense [6,7].

Emissions from biomass burning are generally calculated following the methods described in [8]: total emissions from biomass burning are a product of the Burned Area

(BA), available above-ground Fuel Load (FL) and the Combustion Completeness (CC). Advances in satellite instruments and improvements in detection algorithms have led to global BA datasets becoming ever more accurate: BA detection algorithms have been developed for medium-resolution instruments such as Landsat 8 and Sentinel-2 [9–11], paving the way for a more accurate global BA dataset.

Significant uncertainty remains, however, in estimating emissions from fires on larger spatial/temporal scales due to the difficulty in quantifying FL across larger areas [2,12]. Various studies aimed at calculating fire emissions on larger scales have used different approaches: some estimate available biomass based on look-up tables of regional biome averages, global examples of which include the Fire INventory from NCAR (FINN) [13], but regional studies specifically for savanna fires have also been done in this fashion, such as in [14]. This method has the advantage of potentially covering large areas, but neglects much of the variation within the biome itself. Others use Net Primary Production (NPP) and estimates of turnover rates for various biomass pools to simulate growth, death and other processes that contribute to biomass [15].

Other recent studies have made use of LiDAR data and allometric techniques to estimate Above-Ground Biomass (AGB) [16,17] or image/reflectance data from satellites or other aerial instruments [18,19]. There are trade-offs in resolution/cost for different methods/sensors, and applying a satellite-generated AGB model created at high-resolution (e.g., from Landsat/Sentinel pixels) from one region to another is challenging, unless the biomes are similar and any necessary atmospheric corrections have been done rigorously. Similar spectral signatures in two different biomes may not necessarily be indicative of similar biophysical properties, especially as spectral signatures/vegetation indices often also depend on other factors outside the biophysical properties of the vegetation itself [20]. A previous study attempting to model regional AGB in Southern Africa for various vegetation pools was conducted by [21], whereby absorbed Photosynthetically Active Radiation (aPAR) and Light Use Efficiency (LUE) were used along with other surface-level data (e.g., fractional tree cover) to predict monthly AGB across the entire region. Heterogeneity within landscapes means that these approaches are not straightforward, especially when attempting to model AGB at a higher spatial resolution in this way. On the other hand, attempts to make use of local in situ measurements of the surrounding regions face similar challenges, as field measurements in one area may not necessarily be representative of another.

Unmanned Aircraft Systems (UASs) have already been utilised to address these issues. Image photogrammetry techniques can be used to measure plant canopy, a good indicator of AGB within plant species and functional types across a variety of regions [22]. This approach does not fully address much of the (both living and dead) surface vegetation, which does not extend far above the surface itself; however, this type of vegetation is a major constituent of the fuel consumed in savanna fires. UASs serve as a useful tool in creating remotely-sensed models, as they eliminate the need for atmospheric corrections, and the images they produce have a high spatial resolution, which can be coupled accurately to ground measurements via the use of Ground Control Points (GCPs). These images could also potentially be a first step in scaling up ground measurements to global datasets. The advent of satellite-borne instruments with a spatial resolution in the tens of metres, such as Landsat-8 or Sentinel-2, coupled to field data through UAS images, may help connect field data to space, creating a global dataset of instantaneous biomass.

In this study, we used an UAS with a mounted multi-spectral camera to map areas of vegetation in two regions in Southern African savannas with distinct rainfall rates. We used the measurements to train an instantaneous biomass model based on a combination of features extracted from the remotely-sensed data and meteorological reanalysis data from the ERA-5 Land dataset [23]. The aim was to produce a model capable of predicting AGB in the savanna biome as a whole at a fine scale and in the future to provide a means to link global remotely-sensed datasets from satellites to instantaneous surface biomass.

## 2. Materials and Methods

We used a UAS-mounted multi-spectral camera to map surface reflectances at various locations around the study regions in Southern Africa. The study regions are described in Section 2.1. Ground-level FL measurements were taken at these locations using destructive sampling methods (Section 2.2). We then constructed an AGB model based on a combination of these maps and meteorological data, to simulate surface FL measurements (Section 2.3).

### 2.1. Study Regions

We conducted this study at two sites in Southern Africa. One was at Tsodilo Hills, located in Ngamiland in northwest Botswana (18°44′ S, 21°42′ E). The other was in Niassa Special Reserve, in Niassa Province of northern Mozambique (12°10′ S, 37°33′ E). Both study sites are located in areas classed as "Tropical and Subtropical Grasslands, Savannas and Shrublands" [24]. Tree cover in the Mozambican study region was generally higher than that in Botswana. The locations of the study regions are shown in Figure 1, and examples of the type of vegetation are given in Figure 2.

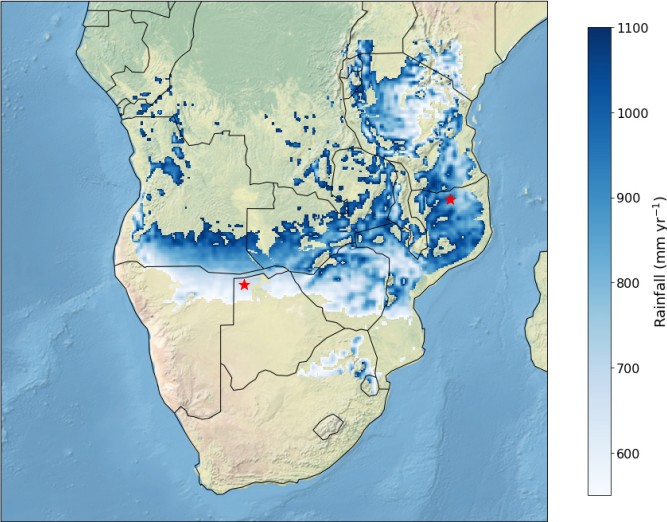

**Figure 1.** Study regions (red stars) and rainfall in areas classified as "Tropical and Subtropical Grasslands, Savannas and Shrublands" [24] falling within the rainfall range of the two study regions (i.e., 550 mm–1100 mm y$^{-1}$ on average over the period 2002–2016).

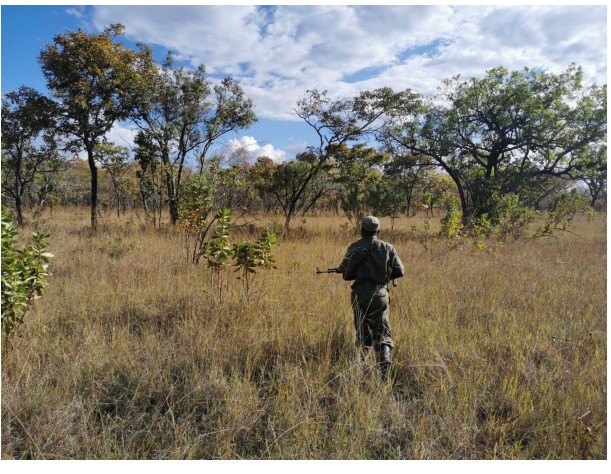
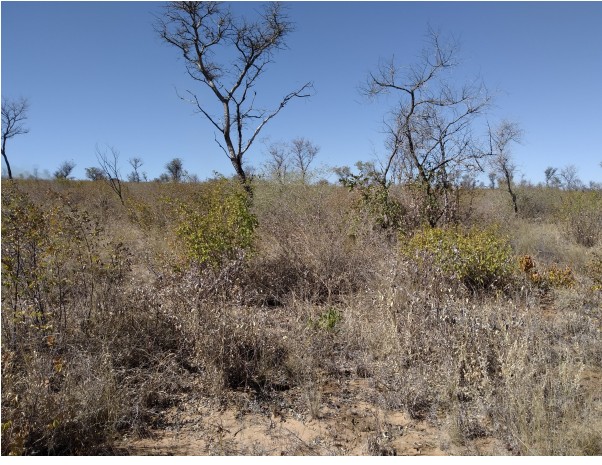

(**a**) Niassa Special Reserve                    (**b**) Tsodilo Hills

**Figure 2.** Example landscapes in the Early Dry Season (EDS) in (**a**) Niassa Special Reserve and (**b**) Tsodilo Hills. Pictures taken by T. Eames and R. Vernooij.

### 2.1.1. Tsodilo Hills

The Tsodilo Hills region is located in the northwestern corner of Botswana, between the Kalahari Desert, the Caprivi strip and the Okavango Delta. In the modern adaptation of the Köppen-Geiger climate classification [25], this area falls under the category "Arid Steppe". The mean annual temperature in the period 2001–2016 was 22.7 °C, and the mean annual precipitation for the same period was 660 mm, as calculated from ERA-5 land dataset. The dry season here runs from April to October, coinciding with the fire season (Figure 3). The majority of the area burned occurs later in the dry season, the fire season starting from July until the rains return in October/November.

This region is characterised by sparse tree cover and relatively high shrub density (common species include *Vachellia Erioloba* and *Combretum collinum*) in a dry savanna landscape [26]. Infertile sandy Kalahari dunes (mostly orientated east-west) dominate, occasionally broken with somewhat more fertile ancient riverbeds, or omuramba. There is a sparse human population spread across a few villages and a number of cattle posts. Areas where these cattle posts, often located around boreholes, which provide water during the dry season, are found are therefore more heavily grazed, and we avoided them as much as was possible while taking ground fuel load measurements. Wildlife presence in the area is severely limited by the lack of water, especially in the dry season.

In this area, we measured 28 plots of 50 m × 10 m (Figure 4b). Eighteen plots in total were measured in the Early Dry Season (EDS) and 10 in the Late Dry Season (LDS). These plots were analysed based on 11 UAS maps, 7 in the EDS and 4 in the LDS. The 2018–2019 wet season rainfall was far below average, about 370 mm recorded by ERA-5. As a result, the landscape had already dried out significantly in the EDS, and even in the earliest plots measured, grasses were already mostly cured. Live woody vegetation was at various stages of deciduousness, with few exceptions. Fractional Tree Cover (FTC) in the study area was low, on average around 4.3% recorded in March 2019 from MODIS FTC data [27].

### 2.1.2. Niassa Special Reserve

Niassa Special Reserve (Reserva Especial do Niassa (REN)) is a protected area in northern Mozambique, bordering Tanzania. It is classed as "Tropical Savanna" and experiences more rainfall than NW Botswana, around 940 mm per year (averaged over 2001–2016). The mean annual temperature is approximately 24.6 °C. The dry season is marginally shorter, May to October. The fire season runs from the start of the dry season, however with a less pronounced skewing to the LDS than in Tsodilo (Figure 3c). We measured 17 plot locations (between 1.5 and 3 ha) in the EDS (June/July) (Figure 4a).

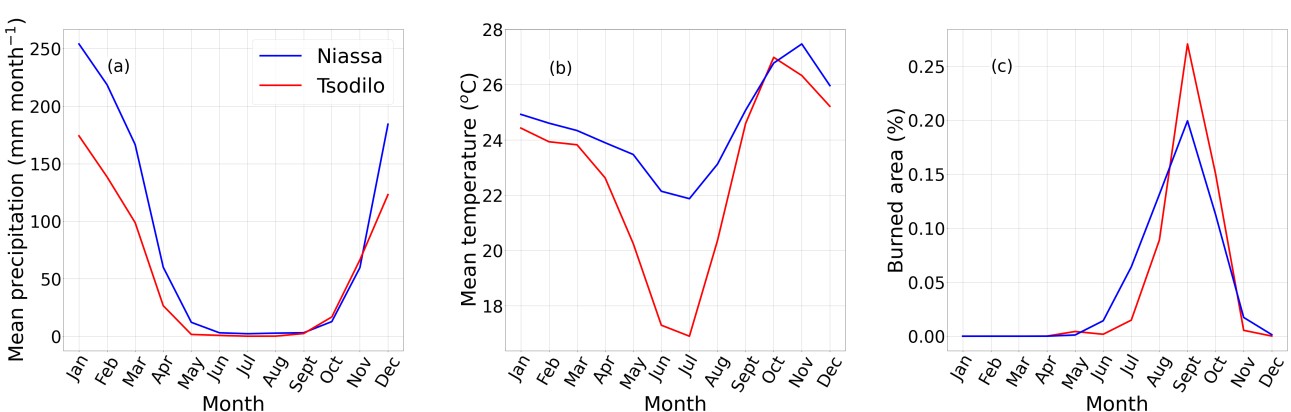

**Figure 3.** Monthly means of rainfall (**a**), temperature (**b**) and percentage of the region burned (**c**) for both study regions in the period from 2001–2016. Precipitation and temperature averages were calculated from the ERA-5 land dataset [23] and burned area from MODIS MCD64A1 C6 [28]. The area from which data were included for these calculations is shown in Figure 4.

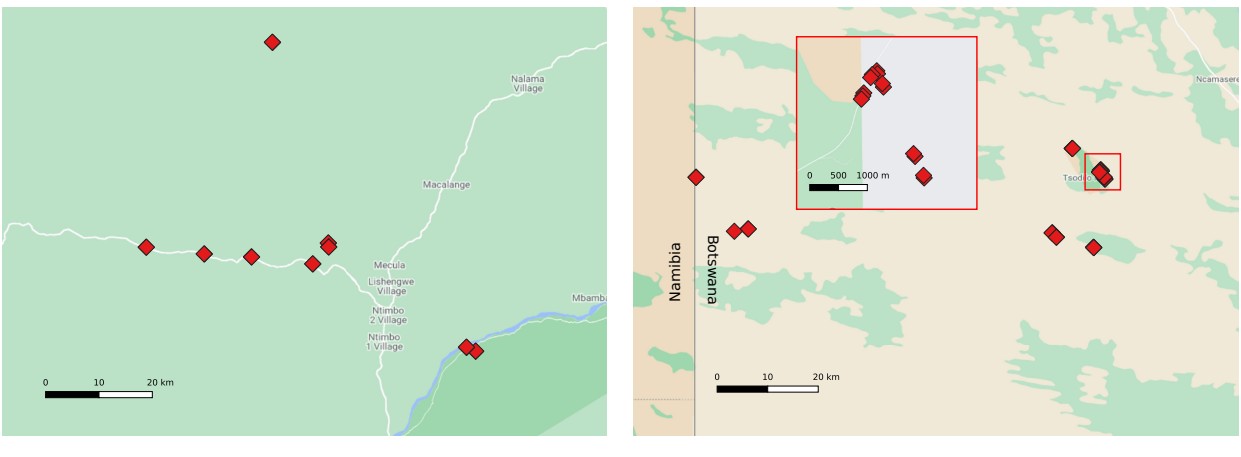

(**a**) Niassa Special Reserve                     (**b**) Tsodilo Hills

**Figure 4.** Plot distribution across study areas in (**a**) Niassa Special Reserve and (**b**) Tsodilo Hills. In all cases except the magnified region in (**b**), the red diamonds represent two or three plots placed in parallel. Within the magnified region, diamonds show single plots.

Much of the reserve falls under the "Miombo" ecoregion, characterised here by relatively undisturbed open woodlands and wooded grasslands. Species such as *Julbernardia globiflora*, *Diplorhynchus condilocarpon* and *Brachystegia boehmii* dominate [29] in these woodlands, which are occasionally broken by rocky inselbergs, (seasonal) dambo wetlands/grasslands and evergreen riverine forests. Most of the soil is sandy/clay, with clay content higher around the Rovuma and Lugenda rivers, which border the reserve to the north and south, respectively. REN is part of the Rovuma River Basin, of which the Lugenda is a sub-basin. These two rivers represent the only perennial water flow in the reserve, though there are a number of smaller seasonal streams [30]. REN is home to a number of species of large herbivores, including elephants, as well as a large (>1000) population of lions [31], African wild dogs and hyenas. The human population within the reserve was roughly 60,000 as of 2019, spread across more than 42 villages and settlements [32]. In laying the sample plots, we endeavoured to avoid areas with any evidence of major human and/or wildlife influence.

Mozambique experienced exceptionally high 2018–19 wet season rainfall, at least in part due to the landfall of two late rainy season tropical cyclones in a relatively short space of time. Niassa Province is further inland and thus was spared the most intense rainfall, though the 2018–2019 summer rainfall was above average, 1045 mm (ERA-5 data). Grasses were green or only just beginning to cure in most plots in the EDS, and in only one plot was a significant portion of the live woody vegetation going deciduous. Fractional tree cover was higher than in Tsodilo Hills, averaging around 28.6% in March 2019.

### 2.2. Ground Measurements

We measured FLs at each site using an adaptation of the methodologies described in [14,33]. Each measurement site consisted of either two or three of the previously described plots in relatively close proximity (around 50m apart), in which we sampled all AGB fuel classes (Table 1). A total of 45 plots were measured, so as to get representative samples of the different vegetation types in the two study regions, capturing areas with varying tree/shrub cover, as well as different fire history. Small ground level fuel (i.e., non-live woody vegetation such as shrubs or trees) was divided into three categories: grass, litter and coarse woody debris. Grass and litter were collected in five 1×1 m squares equidistant along the centre line of the plot, starting at 0 m, 10 m, 20 m, 30 m and 40 m. Coarse fuel was collected in three 5×5 m squares at 0 m, 20 m and 40 m. Individual trees and heavies (logs) were counted along the length of the plot, within 5 m to the left of the centre line and shrubs within 1 m of the centre line to the left. An example plot is shown in Figure 5.

**Table 1.** Summary of the fuel classes measured at the ground level in each plot.

| Fuel Class | Description |
|---|---|
| Grass | All living grasses and dead attached |
| Litter | All dead grass material and living not attached, leaf litter and any woody debris with diameter < 0.6 cm |
| Total fine | Sum of the grass and litter components |
| Coarse | Woody debris with diameter $0.6 \text{ cm} \leq x \leq 5 \text{ cm}$ |
| Heavy | Woody debris with diameter $\geq 5 \text{ cm}$ |
| Shrubs | Live leafy vegetation with Diameter at Breast Height (DBH) < 5 cm and/or height < 2 m |
| Trees | Live leafy vegetation with DBH $\geq$ 5 cm and/or height $\geq$ 2 m |

In each of the squares, all material belonging to each of the classes in Table 1 was collected using destructive sampling methods and weighed directly. Wet weight was recorded using digital scales, and sub-samples were taken from each plot to be oven-dried and re-weighed, to determine moisture content, which then determined the dry weight. The field weights were averaged to give a single value for FL of each category for each plot, in $\text{gm}^{-2}$. A distribution of fuel load across each site is shown in Figure A1 in the Appendix A.

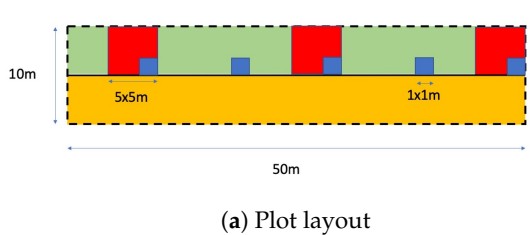

(**a**) Plot layout

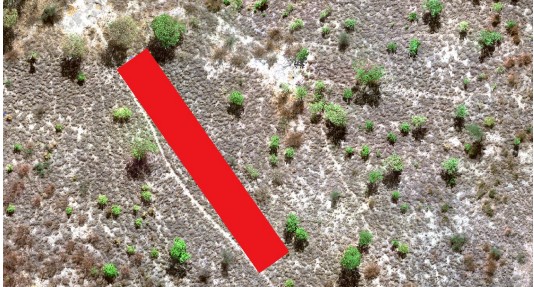

(**b**) Plot within the orthophoto

**Figure 5.** An example individual (**a**) plot schematic and (**b**) location with an UAS-generated orthophoto showing how AGB was measured for the various fuel classes. In (**a**), blue squares show locations where grass and litter were collected, and red squares for coarse woody debris. Heavy fuels were measured in the green half of the plot, shrubs counted within 1m of the centre line in the yellow half and trees counted in the whole yellow section.

### 2.3. UAS Maps

#### 2.3.1. Map Creation

Across both sites, we placed visible targets at the start and end of the plots for direct AGB measurements. We flew a DJI Matrice 100 quadcopter UAS with a mounted MicaSense RedEdge multi-spectral camera (see Appendix B for camera bands) to capture multiple images of each plot. The camera was mounted on a gimbal arm to ensure stability while capturing images. Additionally, the camera was connected to a GPS and Downwelling Light Sensor (DLS) mounted on top of the UAS to assist with georeferencing and correct for changes in light conditions during a flight, though we endeavoured to keep these to a minimum by flying close to midday and in cloud-free conditions where possible. For each set of plots, we stitched a single orthophoto together from multiple multi-spectral images, creating one 5-band orthophoto. Accurate and efficient UAS mapping requires the consideration of several factors, including flight altitude and forward/lateral overlap between images [34]. In order to maintain a high spatial resolution whilst still covering a reasonable area in each flight, we set the flight altitude to 50 m above the ground level to obtain an image resolution of about 5 cm, which would allow for reasonable spatial coverage while still enabling the distinction of much of the small-scale fuel burned.

Orthophoto mapping was pre-planned on-site to cover as much of the same vegetation type as possible. We used DJI GroundStation Pro software to create flight plans and adjust

flight variables such as flight speed or the image acquisition interval. These were set such that every image had a consistent forward and lateral overlap of 80%, to optimise image overlap and improve stitching. Area covered by each flight varied from about 1.5–3.5 hectares. We ensured at least 2 Ground Control Points (GCPs) were visible by the camera at either end of all plots, to identify the plot and also for later use in georeferencing the stitched image, to improve geolocation accuracy.

2.3.2. Map Processing

The output of a single flight was a series of images, each subdivided into 5 bands. A slight misalignment between the bands was corrected using the Python libraries provided by MicaSense (found here: https://github.com/micasense/imageprocessing). Following this, orthophoto stitching and Digital Elevation Map (DEM) creation was done using Open Drone Map (ODM), an open-source mapping software for UAS imagery and a viable alternative to commercial software such as Pix4D or Agisoft PhotoScan [35,36]. Two Vegetation Indices (VIs) were calculated from the orthophoto to aid with image classification. We chose the Normalized Difference Vegetation Index (NDVI) [37,38] to discriminate between land cover types (e.g., bare soil and vegetation) [39,40] and the Burned Area Index (BAI) [41] for better classification of shadows, as well as burned area [42,43]. These VIs were added to the orthophoto as extra bands, so that every pixel in the orthophoto had 8 "features"; NDVI, BAI, elevation (DEM) and (digital number) reflectance in all 5 camera bands (Appendix B) illustrated in Figure 6.

We then classified the stitched images into 5 different categories using Object-Based Image Analysis (OBIA) in Python [44]. The number of individual "tree-like" objects (including both trees and shrubs) was counted in every plot using an implementation of Mask R-CNN [45] complemented with the counting of individual "foliage" objects in the classifier (Figure 7). We isolated each plot in turn from the relevant orthophoto and classified and calculated mean plot features, taking care to exclude areas covered by shadow. This was done using first a handheld GPS (accurate to 3 m) and, following this, manual corrections to major misalignments by visual inspection. It was assumed that the (averaged) ground measurements were representative of the plot as a whole, so we extracted features for the full $50\times10$ m area.

For meteorological and climate variables, we used data from the grid point nearest the plot from ERA-5 and added them to the feature list (Table 2). These variables were chosen as they represent important (limiting) factors in vegetation growth (e.g., available water and sunlight), as well as influencing the production of dead material (litter and coarse), both indirectly through biomass production and directly through climatic drivers of litterfall [46–49]. Additionally, we chose Time Since Fire (TSF), defined as the amount of time that has passed since the last fire registered in the MODIS burned area product [28]. This was included on the basis that savanna fires predominantly consume surface fuels, and therefore, a fire would "reset" the surface-level biomass in the area. This is not always the case, and in a number of plots in Niassa, some of the moister grasses were observed to survive fires in the EDS. This variable was also used to define the time span over which to sum or average the meteorological variables.

The final step was to feed these features into an Ordinary Least Squares (OLS) regression, to calculate coefficients for these features that could be used to predict FL. This was done individually for each fuel class. OLS was chosen to maintain simplicity, and after inter-comparison with other models (e.g., Python implementations of Ridge/ElasticNet [50]), it also showed its predictive power to be on par with these models or better. Due to the sparsity of the data, to achieve more robust results, the regression was run multiple times, whereby each time, a single data point was used as a "test" to assess the accuracy of the model, similar to the method of "Leave-One-Out Cross-Validation" (LOOCV). Averages were taken of the results above a given accuracy threshold to calculate a single coefficient for each feature and produce a linear model for surface FL for all fuel categories.

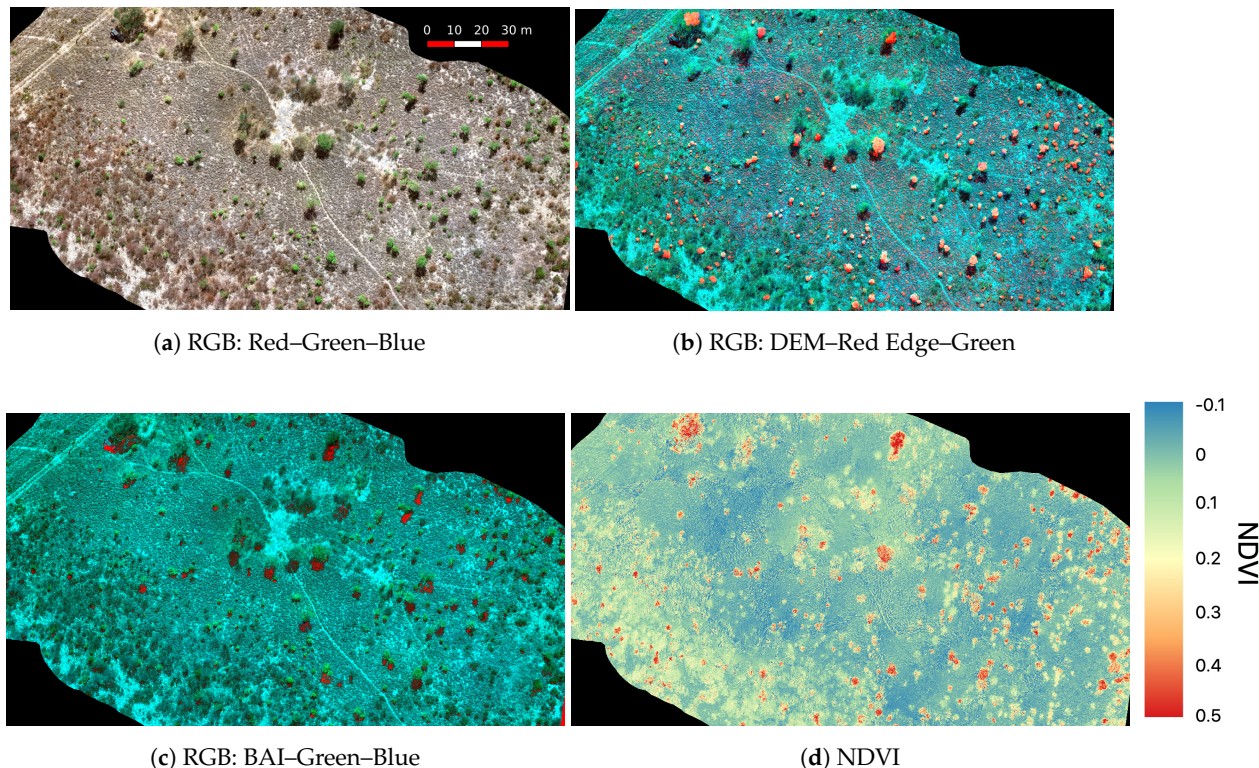

(**a**) RGB: Red–Green–Blue          (**b**) RGB: DEM–Red Edge–Green

(**c**) RGB: BAI–Green–Blue          (**d**) NDVI

**Figure 6.** (False-)Colour composites (**a**–**c**) and a single-band pseudo-colour image (**d**) from an orthophoto of an example plot in Tsodilo Hills, Botswana: (**a**) regular RGB image; (**b**) canopy height; (**c**) shadows; and (**d**) vegetation greenness. The total area of the plot displayed here is approximately 2.8 ha.

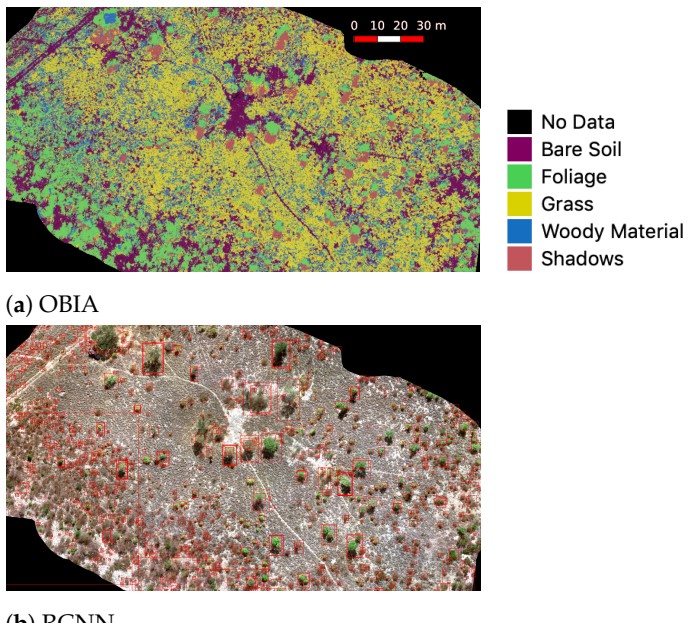

(**a**) OBIA

- No Data
- Bare Soil
- Foliage
- Grass
- Woody Material
- Shadows

(**b**) RCNN

**Figure 7.** Example classification done using OBIA, where (**a**) shows the classification and (**b**) tree-like objects detected by the Mask R-CNN model.

Each fuel class had a total of 45 data points. During modelling, seven were selected as a testing subset and withheld and training was performed on the remaining 38. The

model training output is presented in Appendix C, and the predictive power of the model is discussed in the following section.

**Table 2.** Features used in modelling the FL. A single, mean value for every feature was computed for each plot. Averages and sums of meteorological variables from ERA-5 were taken spanning from the date of the last previous fire detected at a given location and the date the measurements were taken.

| Feature | Source |
|---|---|
| TSF (days) | MCD64A1 C6 |
| Proportion of foliage (% area) | UAS classifier |
| Proportion of grass (% area) | UAS classifier |
| Proportion of bare soil (% area) | UAS classifier |
| Tree/shrub number | UAS image and Mask R-CNN |
| Total precipitation (m) | ERA-5 monthly $0.1°\times0.1°$ |
| Mean temperature (K) | ERA-5 monthly $0.1°\times0.1°$ |
| Mean surface net solar radiation ($Jm^{-2}$) | ERA-5 monthly $0.1°\times0.1°$ |
| Total evaporation (m of water equivalent) | ERA-5 monthly $0.1°\times0.1°$ |
| Mean soil moisture ($kg\ kg^{-1}$ in the upper soil layer, 0–7 cm) | ERA-5 monthly $0.1°\times0.1°$ |

## 3. Results

Before the presentation of the model results, it must be re-stated that the number of data points available was low (38 training and seven testing). It is therefore difficult to draw robust conclusions. Nonetheless, we believe that these results are promising and will provide a solid foundation for future work.

The model represented the field measurements reasonably well in the fine fuel categories, and grass had the highest $R^2$ of any fuel category (Table 3). Full results across all fuel categories are shown in Figure 8. The heavy woody debris category was excluded from the analysis, as the modelled AGB in this category was challenging (on-site weighing of large items is problematic), and very often, heavy fuels did not burn significantly in the fires observed. Coarse woody debris was the worst performing class, but was kept in the analysis as this class is more relevant in savanna fires. Difficulty in modelling AGB in this fuel class likely has much to do with this type of fuel type being difficult to detect directly (unlike grass, trees or shrubs). Coarse woody material is also less likely to fall from trees/shrubs regularly in the same way as leaf litter would, but would fall as a result of irregular activity such as interactions with animals, extreme weather events or tree mortality and decay. However, coarse debris constitutes a small percentage of the total surface FL (Figure 9), and we found that the total fine class indeed has the best percentage accuracy score and smallest percentage error range of any class (Table 3). This would suggest that surface fuels that are predominantly consumed in savanna fires are also the best represented by the model.

**Table 3.** Model prediction Root Mean Squared Error (RMSE), mean percentage error and the *p*-value for different fuel classes, with corresponding error ranges where relevant. Percentage error was calculated as a percentage of the measured value; in the instances where the measured value was zero, it was set to one to avoid zero division.

| Fuel Class | Model $R^2$ | RMSE (min, max) | Mean % Error (min, max) | *p*-Value |
|---|---|---|---|---|
| Grass | 0.913 | 133.95 (6.13, 225.84) | 105.26 (2.41, 349.94) | 0.0007 |
| Litter | 0.867 | 71.44 (18.78, 109.07) | 158.14 (7.74, 909.88) | 0.002 |
| Total Fine | 0.769 | 111.53 (1.27, 184.09) | 23.09 (0.33, 59.19) | 0.009 |
| Coarse | 0.255 | 42.57 (4.14, 99.56) | 471.19 (7.80, 2048.56) | 0.009 |
| Shrubs | 0.513 | 13.89 (0.00, 28.00) | 73.03 (0.00, 300.00) | 0.007 |
| Trees | 0.807 | 1.81 (1.00, 3.00) | 67.38 (16.67, 100.00) | 0.006 |

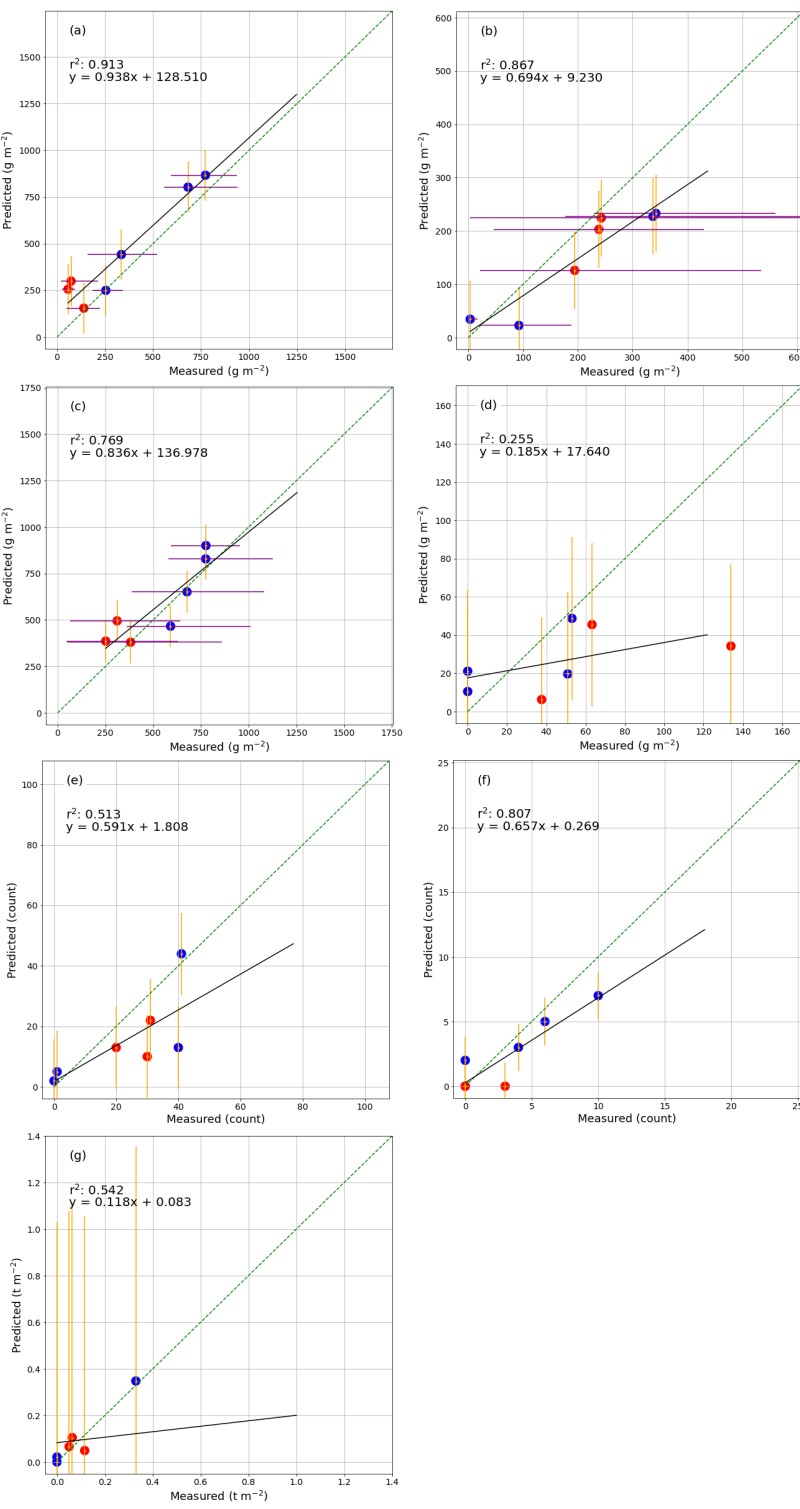

**Figure 8.** Predicted AGB compared to actual measured AGB values for (**a**) grass, (**b**) litter, (**c**) total fine fuel, (**d**) coarse, (**e**) shrubs, (**f**) trees and (**g**) heavy. Plots in Botswana are shown in red and Niassa in blue. In all plots, orange error bars show the extent of the RMSE error of the model in each fuel class. In some classes where data were collected at multiple locations and then averaged (grass, litter and, consequently, total fine fuel), purple horizontal lines indicate the range of values measured within individual plots, and the points then indicate the mean.

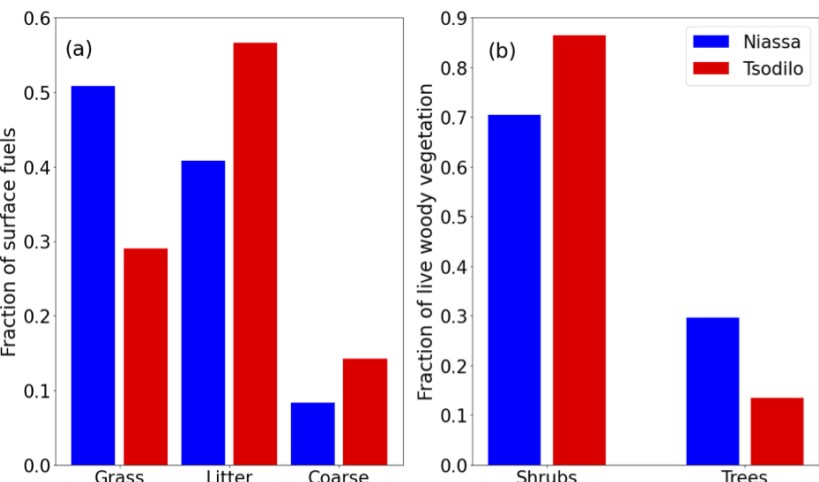

**Figure 9.** Mean proportion of ground measurements represented by (**a**) each surface fuel class to total surface fuels (the sum of grass+litter+coarse) and (**b**) live woody fuel classes (i.e., live trees or shrubs) to total live woody vegetation (the sum of trees+shrubs) for the two study regions.

## 4. Discussion

### 4.1. Model Output

In general, prediction errors were smaller for surface fuel classes, and the total fine fuel class showed significantly smaller percentage errors than other classes. The model tended to show positive bias in low biomass regions and vice versa in high biomass regions, across all fuel classes and both study regions (Figure 10). This would indicate perhaps that separate models may need to be developed for "low" and "high" AGB areas, which may also be reflected in the different regions; it would be an interesting extension to this study to create separate models and investigate the reasons for disparities among them. In this instance, however, there were insufficient data to perform a robust enough investigation of those effects.

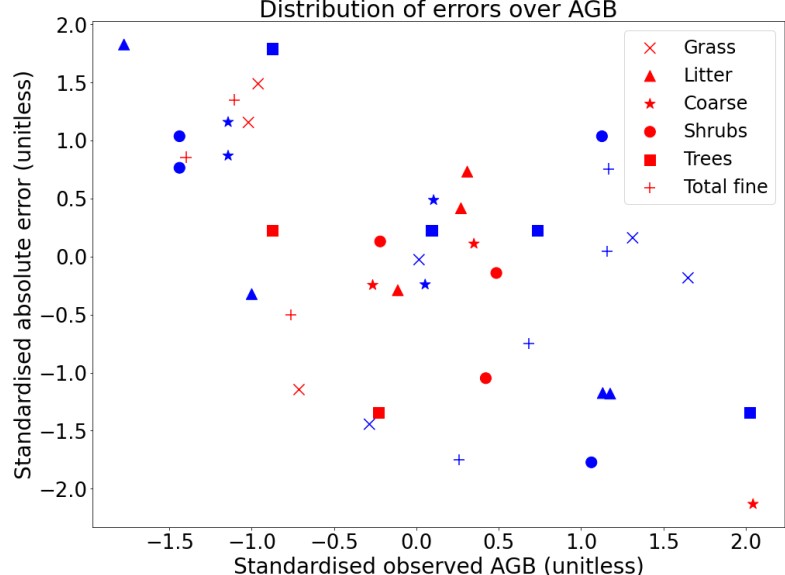

**Figure 10.** Standardised error on observed values across all fuel classes and regions. A red marker indicates data from Tsodilo and blue from Niassa.

For grass as well, the range of values measured on the ground in each 1×1 m quadrant was well represented by the model error range in this fuel class (Figure 8a). Litter measurements were much more variable within each plot (Figure 8b). Grass loading in such a small

region tends to be more uniform (with the possible exception of heavily grazed areas), and litter loading in each quadrant depends heavily on the placement of the transect; should one quadrant fall precisely under a tree, then the litter loading is high, and in between two trees, then the measured value is likely to be low despite likely high loading in the surrounding area. Each plot was chosen to be a good representation of vegetation in the area, and as such, despite the large ranges measured, we think the averaged values show a reasonable representation of litter AGB in a single plot. The internal variability does present a challenge for accurate modelling however, and this effect was present (though damped by the relatively more uniform grass measurements) in the total fine class. It should also be noted that the ranges in the total fine class were calculated by adding the two lowest and two highest grass and litter loadings together, respectively, but this is not necessarily realistic. In all likelihood, areas with the highest grass loading will have the lowest litter load (pure grasslands) and vice versa for highest litter load (pure shrubland/woodland).

Noteworthy as well is that for trees, the Botswanan plots were predicted exclusively at zero. For trees, the OLS fit was far stronger when the two regions were viewed together rather than separately (which can be seen in the respective spreads of blue and especially red points in Figure A2f). For total fine fuel in Botswana as well, the trend was not immediately obvious when blue and red points were considered separately. Perhaps, this is partially due to generally lower fine fuel loading in Tsodilo compared to Niassa, particularly for grass. Additionally, the Botswanan study site included substantial areas where the landscape is impacted by cattle and goats, the grazing of which acts as an external influence on the contribution of grass load. Efforts were made to avoid areas where cattle were known to graze intensively, but this factor cannot be ruled out completely.

Some factors played a greater role than others in the predicted value of AGB (Figure 11). The model for grass AGB relies on a combination of factors, most strongly the proportion of grass pixels detected by the UAS, but soil moisture, solar radiation, number of tree-like objects and temperature also play a role. For the litter class, the number of tree-like objects showed the greatest explanatory power, closely followed by % grass-covered area. The relationship between grass and litter is a complex one, and it varies between species and biome. In this instance, it was not necessarily directly causal. It is more likely that simply in areas with higher tree/shrub density and, by extension more twig/leaf litter producing species, there is generally less grass cover, and vice versa, possibly a result of less light penetration and reduced understorey growth. The tree-like object count, however, was likely to be causal, as the more individual litter-producing plants there are, the more surface litter there would be.

The explained variance of the total fine fuel class is largely dominated by the same features as grass, which would be again logical if grass contributed the most to total fine fuel AGB. This is the case for plots in Niassa, but not for those in the Tsodilo region (Figure 9a). This explains to some extent why the fit for the predicted surface biomass in Tsodilo showed less of a clear trend (Figure A2c) than that in Niassa, as the distribution of AGB across surface fuel classes differed and was skewed more towards litter in Tsodilo and grass in Niassa. Total AGB was generally lower in Tsodilo as well, so these data points perhaps contributed less to the fitting error, making them "less important" in a fitting context and potentially resulting in a total fine fuel model slightly better suited to more moist, grassy areas of savanna.

Interestingly, soil moisture was the most important explanatory variable for the number of trees in a plot, but not for the number of shrubs. This discrepancy may be a result of regular fires consuming smaller shrubs and not trees in regions like Niassa where almost all the plots burn annually, but could also be due to the difference in soil moisture between the regions. As previously mentioned, the fit in the trees class was weaker when regions were considered individually, and almost all plots in Niassa registered more trees (~8 on average) per plot than those in Tsodilo (~2 per plot). Since Niassa is also a warmer, wetter region in general, a correlation was found between the overall trend in tree numbers and soil moisture and used as an explanatory variable, but given that the same was not

found for shrubs, it seems more likely that this was an artefact of the model design/chosen study regions. A similar effect may be observed for solar radiation and temperature, as in general, plots in Tsodilo experience a cooler climate (Figure 3b) and somewhat more intense mean downward solar radiation (21.9 MJ m$^{-2}$) than Niassa (19.8 MJ m$^{-2}$) in ERA-5 data.

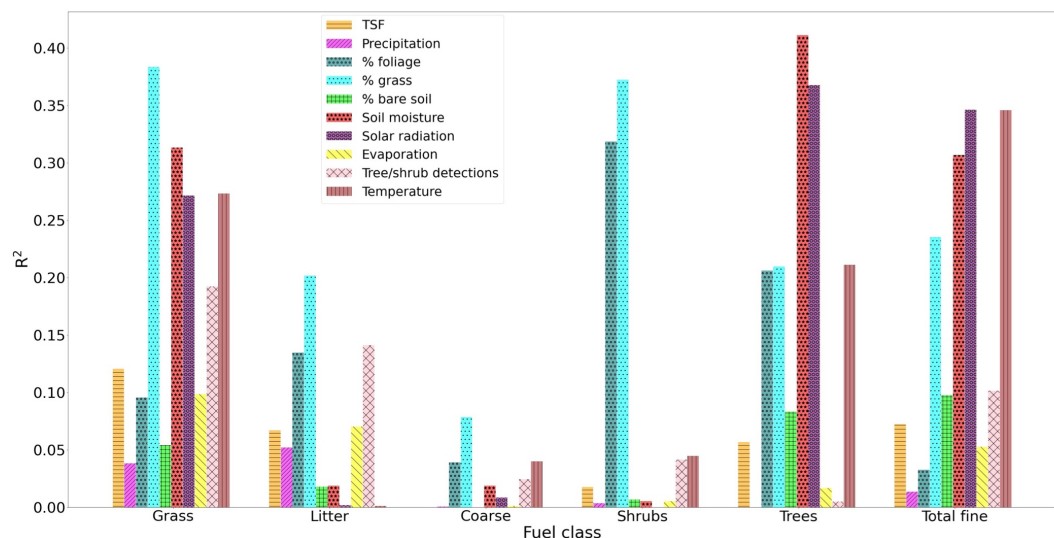

**Figure 11.** The explained variance of measured AGB due to each feature used in the model. TSF: Time Since Fire.

In most cases, a combination of factors, from the UAS and from meteorological input, appeared to contribute a similar proportion of the total R$^2$. One exception was in the shrubs category, where only two features (% grass and % foliage) explained significantly more of the variance than the others, both of which were extracted from UAS data. There may also be some overlap in the explained variance as a result of the proportions of foliage, grass and bare soil, as if there was a higher percentage of pixels in a given plot classed as "grass", then there must be a smaller percentage classed as foliage or bare soil. This effect is likely to manifest itself in other variables as well. Soil moisture is also affected by temperature, evaporation, precipitation or solar radiation, and indeed, the magnitude of the explained variance for soil moisture (and to a lesser extent, temperature) and solar radiation was similar and opposite (i.e., when one was proportional, the other was inversely proportional) in every class where these features made a relatively more significant (R$^2$ > 0.2) contribution. Separate models tailored to individual vegetation classes could be a further improvement in this regard, especially if the same accuracy could be achieved with fewer input variables.

### 4.2. Upscaling

The bounding box used to delineate plots within the model training data was 500 m$^2$ (22 m×22 m), approximately the same scale as the more recent remote sensing instruments found on LANDSAT/Sentinel-2 satellites, showing potential for the upscaling of these measurements to satellite products. An example of the grass AGB model applied to a plot in Tsodilo Hills is shown in Figure 12. To achieve this, five-hundred meters squared tiles were delineated in the orthophoto, and the biomass for the surface fuel classes was calculated for each pixel. This dataset (along with relevant errors) was then re-gridded to the Sentinel-2 grid. With this, we can create a dataset of UAS predicted AGB, on the scale of satellite images, to act as a "truth" dataset. This dataset can be used to train a machine learning model in order to predict AGB on the basis of satellite imagery.

Direct reflectance comparison between satellite instruments and UAS data has many pitfalls (exact date/time of retrieval, atmospheric corrections/distortions, differences in scale, divergences in band definitions/widths/response curves, etc.). In this study as well, no particular effort was made to calibrate measurements with satellite overpasses, and the relevant bands in some cases barely overlapped (see Table A1). One of the advantages

of the approach detailed in this paper is that the need for reflectance data comparison was circumvented through the use of classification maps and surface cover proportions. Another potential way of connecting UAS-generated AGB maps, alternative to machine learning, would be to generate sub-pixel surface cover percentages from satellite images and feed these into the model directly in place of the UAS-generated surface cover. This approach, and the machine learning approach mentioned above, will be the subject of further research.

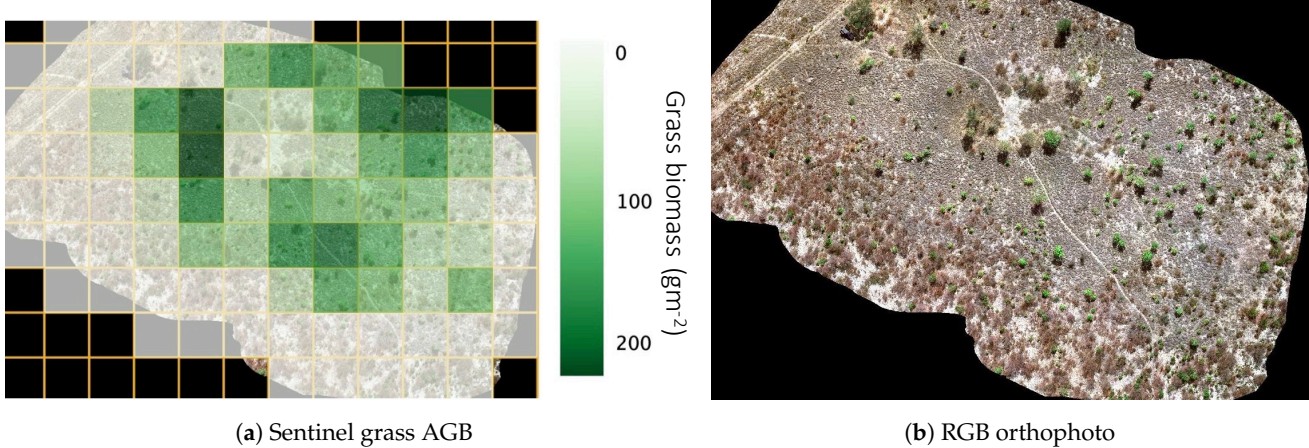

(**a**) Sentinel grass AGB        (**b**) RGB orthophoto

**Figure 12.** Example of (**a**) the grass AGB model applied to Sentinel-2 pixels in a plot in Tsodilo and (**b**) and RGB image of the plot for reference. Each box in the yellow grid in (**a**) corresponds to a Sentinel-2 pixel.

## 5. Conclusions

We developed a fuel load model based on UAS measurements and ancillary data using matching measurements in the field as calibration data. The model was able to reproduce AGB well for most fuel classes, especially for grass and surface fine fuels, though not without (occasionally large) errors. The total fine fuel class, perhaps the most relevant class for savanna fires, has the smallest percentage prediction error. We can therefore apply this model to tiles around the size of a Landsat/Sentinel-2 pixel with a reasonable degree of confidence for fine fuel classes and obtain good estimates for surface-level biomass on a larger scale. As the regions studied include a range of savanna types, arid to wet with a range of tree covers, the model may be applicable to a large range of savanna landscapes in Southern Africa (see Figure 1).

We showed that it is possible to achieve instantaneous remotely-sensed biomass estimates within savanna biomes in a relatively simple way, using freely available meteorological data in combination with UAS maps made from open-source software. The methods described in this paper work best for surface fuels, so in the context of biomass burning, they are most useful when considering surface and/or low-severity fires. Further insight may be gained into the fire mechanics in future research by considering the separate pools of surface biomass and better encompass crown fires by improving on the detection of trees, shrubs and the biomass thereof. LiDAR data would potentially be a valuable addition here. Nonetheless, it is a promising step in the direction of a large-scale remotely-sensed near real-time biomass dataset.

**Author Contributions:** Conceptualization and methodology, T.E., R.V. and G.R.v.d.W.; formal analysis, T.E.; investigation T.E., J.R.-S., C.Y., A.E., F.S., N.R. and R.V.; writing, original draft preparation, T.E.; writing, review and editing, T.E., J.R.-S., C.Y., A.E., F.S., N.R. and G.R.v.d.W.; visualization, T.E.; supervision, G.R.v.d.W.; funding acquisition, G.R.v.d.W. and J.R.-S. All authors read and agreed to the published version of the manuscript.

**Funding:** This work was funded by grants from KNAW AMMODO and the Netherlands Organisation for Scientific Research (NWO), and contributions to fieldwork costs were provided by Australian Government funding.

**Institutional Review Board Statement:** Not applicable.

**Informed Consent Statement:** Not applicable.

**Data Availability Statement:** Data available on request.

**Acknowledgments:** The authors wish to thank the Tsodilo Hills Community Trust and Niassa Special Reserve for their hospitality, assistance and support in the fieldwork campaigns. Thanks are also extended to Robin Beatty and Shaun de Lange for their help and advice during the Botswanan portion of the field campaign.

**Conflicts of Interest:** The authors declare no conflict of interest. The funders had no role in the design of the study; in the collection, analyses or interpretation of data; in the writing of the manuscript; nor in the decision to publish the results.

## Abbreviations

The following abbreviations are used in this manuscript:

| | |
|---|---|
| AGB | Above-Ground Biomass |
| aPAR | Absorbed Photosynthetically Active Radiation |
| BA | Burned Area |
| BAI | Burned-Area Index |
| CC | Combustion Completeness |
| DEM | Digital Elevation Model |
| DLS | Downwelling Light Sensor |
| DN | Digital Number |
| EDS | Early Dry Season |
| FL | Fuel Load |
| FTC | Fractional Tree Cover |
| GCP | Ground Control Point |
| GPS | Global Positioning System |
| LDS | Late Dry Season |
| LiDAR | Light Detection and Ranging |
| LOOCV | Leave-One-Out Cross-Validation |
| LUE | Light Use Efficiency |
| NCAR | National Center for Atmospheric Research |
| NDVI | Normalized Difference Vegetation Index |
| NPP | Net Primary Production |
| OBIA | Object-Based Image Analysis |
| R-CNN | Regional Convolutional Neural Network |
| REN | Niassa Special Reserve (Reserva Especial do Niassa) |
| TSF | Time Since Fire |
| UAS | Unmanned Aircraft System |

## Appendix A. AGB Distribution per Site

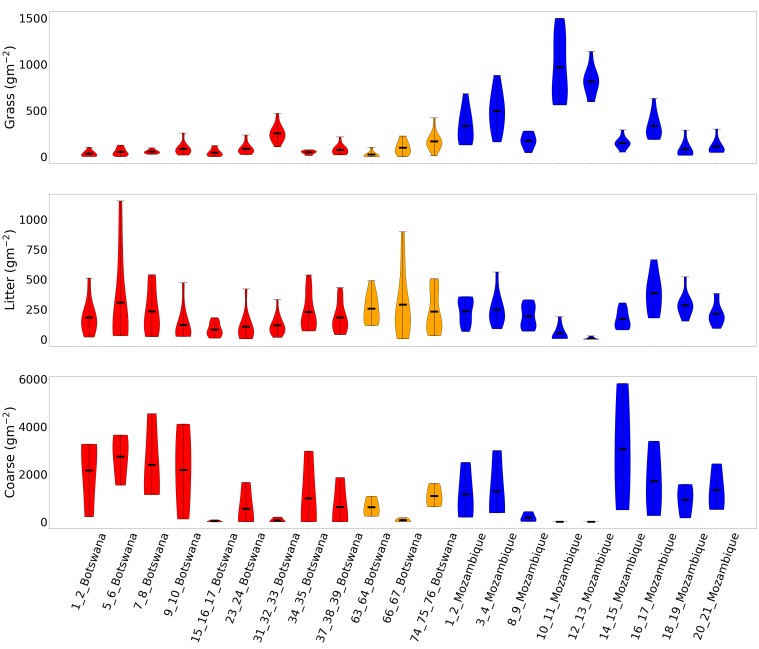

**Figure A1.** Site distribution of fuel load across grass, litter and coarse categories. Red indicates a site in Tsodilo in the EDS, orange in Tsodilo in the LDS, and blue a site in Niassa.

## Appendix B. MicaSense RedEdge Bands

**Table A1.** MicaSense Red edge camera bands and corresponding satellite instrument bands.

| Band Name | MicaSense | LANDSAT OLI | Sentinel-2 MSI |
| :---: | :---: | :---: | :---: |
| Blue | 465–485 | 452–512 | 458–523 |
| Green | 550–570 | 533–590 | 543–578 |
| Red | 663–673 | 636–673 | 650–680 |
| Red edge | 712–722 | N/A | 733–748 |
| NIR | 820–860 | 851–879 | 855–875 |

## Appendix C. Model Training

**Table A2.** Model train parameters.

| Fuel Class | $R^2$ | RMSE | Condition Number |
| :---: | :---: | :---: | :---: |
| Grass | 0.88 | 309 | $8.2 \times 10^9$ |
| Litter | 0.91 | 208 | $8.3 \times 10^9$ |
| Total fine | 0.94 | 433 | $8.3 \times 10^9$ |
| Coarse | 0.70 | 55 | $8.3 \times 10^9$ |
| Shrubs | 0.85 | 34 | $8.1 \times 10^9$ |
| Trees | 0.83 | 6.7 | $7.9 \times 10^9$ |
| Heavy | 0.66 | 0.34 | $8.3 \times 10^9$ |

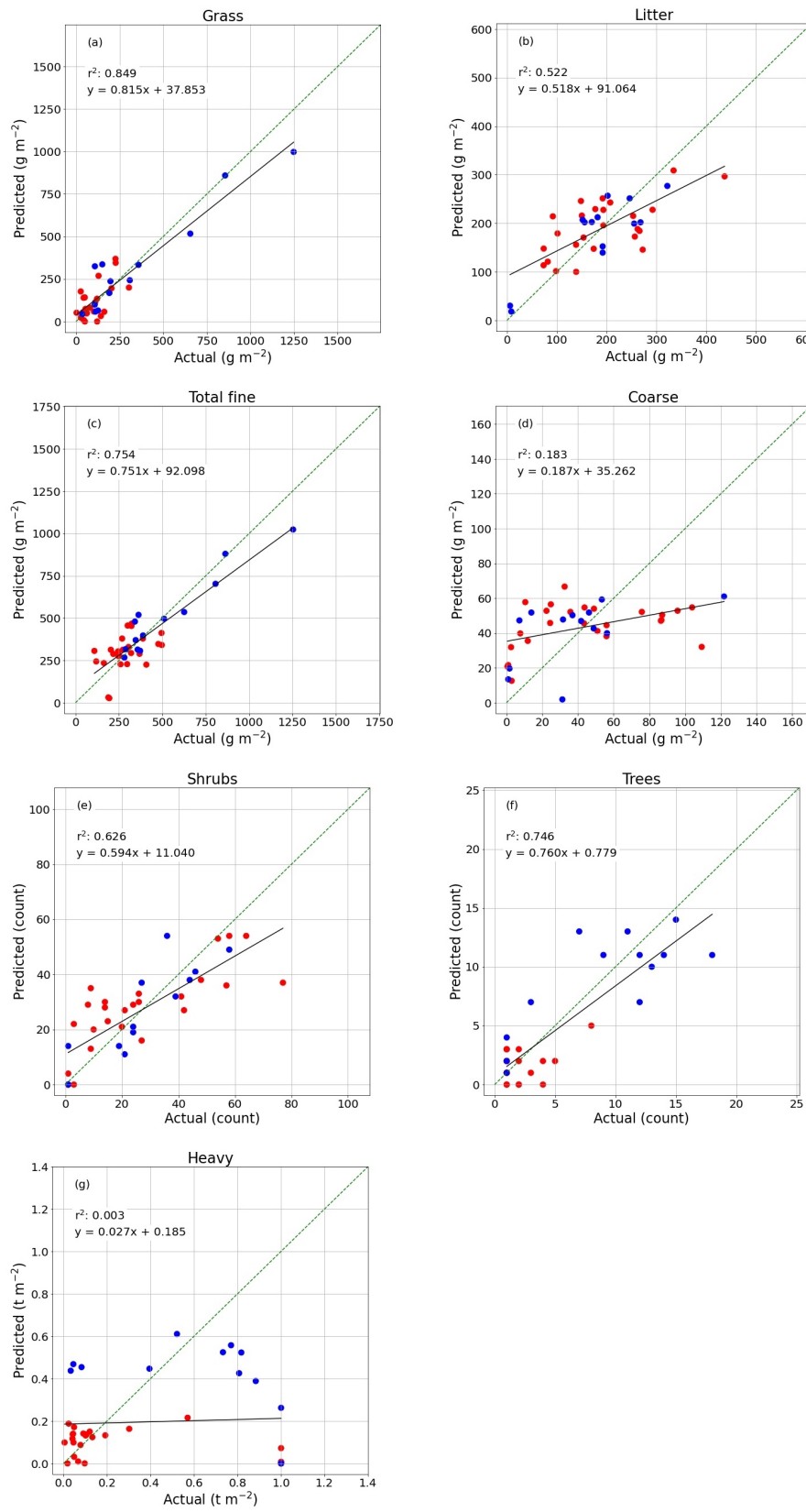

**Figure A2.** Training data values for (**a**) grass, (**b**) litter, (**c**) total fine fuel, (**d**) coarse, (**e**) shrubs, (**f**) trees and (**g**) heavy. Plots in Botswana are shown in red and Niassa in blue.

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
