# Peer review of "Instantaneous Pre-Fire Biomass and Fuel Load Measurements from Multi-Spectral UAS Mapping in Southern African Savannas"

_fire, doi:10.3390/fire4010002_

Round 1
Reviewer 1 Report
I reviewed this paper in its previous version and am pleased to see that the authors have made recommended improvements. I think this manuscript now is a solid contribution to this journal and I recommend publication in its present form.
The new figure showing the study site locations in a better context of similar vegetation and climatic conditions is very helpful, as was the addition of new material relating to upscaling this work to larger areas.
Author Response
We thank the reviewer for his or her (re)reading of our manuscript, and for the useful comments and feedback!
Reviewer 2 Report
This paper illustrates two application of pre-fire and fuel load measurements to understand the impact of landscape fires on greenhouse gases and aerosol emissions. It thus address an important and useful topic in the framework of the climate change mitigation strategy. The choice of two different areas of study is much appreciated, however a more accurate description of the both landscapes and local situation is needed: i.e vegetation types (with photos), presence/absence of local fauna, presence/absence of settlements, hydrografic network, soil type, link to production systems. The integration of several technics to retrieve information is well appreciated as it represents a clear added value in the study. The measurement method could be better described, i.e. time and people involved, method of sampling. The UAV’s description of specifications should be improved, not relying only on references and on the table of spectral bands. The upscaling methodology is promising but too briefly addressed. I suggest to clarify that it is only a tentative approach that needs deep investigations in further studies.
It is not clear what is the extension of the single area of study with the measurements plots inside. A Figure with the two areas and dots of plots would help.
Some photos of the typical vegetation would be useful.
Line 315: replace “is it” with “it is”
Reviewer 3 Report
The manuscript describes the application of UAV imagery to characterise the vegetation structure at sites in Botswana and Mozambique in southern Africa. These data and climate\meteorological data are used to develop a model in order to estimate of above ground biomass for different fuel classes (e.g. grass, litter, downed woody material, trees). The results indicate that the approach has potential. Comparison between field measurements and modelled estimates indicates fine fuels are estimated with the least uncertainty and these estimates with scaled up to Sentinel-2 spatial resolution (~20x20m).
Fuel load is an area of uncertainty in the fire science community and this manuscript describes an approach that exploits the spatial resolution of UAV data to help address it. The manuscript falls within the remits if the journal and it is suitable for publication with minor edits. A list of comments is detailed below.
Specific Comments
#22 – include the reference for southern hemisphere Africa wildfire emissions.
#61 - this sentence would benefit from being rephrased “This approach does not fully address much of the (both living and dead) surface vegetation…”. Is it referring to the separation the living and dry material?
#98 – were the same plots measured at the start and end of the dry season or was each plot only measured once (either start or end)? How were the plots identified? It would be useful to provide a figure for each site that indicates the spatial distribution of plots within the study areas. Also, to avoid confusion with the FL sampling locations (referred to as plots on #121) these could be referred to as (e.g.) measurements sites or similar.
#129 – a figure\table of the range of the measured FL values for each category at each site would be useful.
#163 – The classification makes no distinction between green and senescent foliage whereas the fuel classes appear to have live (grass) and dead (litter) material. How the trees and shrubs were separated from one another (- using the height metrics in Table 1)?
#175 – “Additionally…. biomass in the area”. This sentence would benefit to being rephrased. I assume this is the time period between the last fire occurrence and the point of measurement over which the fine fuels accumulate (assuming that the fire removed all fine fuels)?
#194 – whilst this is might be the case, it would be better state it later in the manuscript
#213 - some field plots were measured either at the start or end of the dry season. Does the temporal difference (changes in FL and meteorology) impact model development?
#267 – what is the fire return interval for these sites? TSF doesn’t appear to have much influence (Figure 9) on any of the fuel class – is fire occurrence similar between sites?
#290 – the approach in scaling up the AGB is a little unclear. Were the AGB estimates for grass scaled to 22x22m resolution on the basis of %grass cover in these cells? If this is the case, how would the AGB be extrapolated over an (e.g.) Sentinel-2 image? Via classification?
#296 – although this is a first attempt at scaling up the AGB estimates, it would be useful to assess how they compare with independent estimates
#297 – is the reference to ‘direct comparison’ between satellite and UAV data with reference to methods of scaling up AGB estimates to a wider spatial extent?
Figure 2 – what is the spatial extent over with the mean meteorological parameters and % burned area are retrieved over each site?
Figure 4 – the figure caption could be rephrased to highlight the scene features which the images \ composites are designed to extract (such as vegetation greenness, canopy height, shadow etc) and in the titles adding (e.g. RGB : DEM-RedEdge-Green)
Figure 4/5 – it would be useful to include a scale bar or include the area covered by the orthophoto
Author Response
Please see the attachment.

This manuscript is a resubmission of an earlier submission. The following is a list of the peer review reports and author responses from that submission.
Round 1
Reviewer 1 Report
The paper addresses an important topic. AGB estimation based on remote sensing is still characterized by large uncertainties, especially when considering larger scales. Therefore, any improvement that could lead to a better estimate on pre-fire fuel load is central to reducing the uncertainties on GHG emissions from fires.
I started to read the paper with high expectations as I new of some of the authors and their work. However, at the end I felt a bit disappointed as it presented just a small improvement in pre-fire fuel load estimation. It is neither very innovative nor is its applicability to larger scales useful. It is a step further but deserves more research. Nevertheless, the work is of value and based on solid scientific methods and I recommend its publication.
Below are my general comments followed by some specific remarks:
- The authors present fuel load as the largest source of uncertainty of the equation to calculate fire emissions using the area approach. However, I disagree as combustion compleness is still the largest source of uncertainty. Fuel load has had a lot of field research resulting in many published estimates. The problem is how to account for spatial variability within the same land cover class and during the season, and how could this be upscaled.
- The title refers to an “instantaneous” measurement. This implied user-ready applicability of the model, and therefore that the same model can be applied in other areas without extracting new parameters. This paper does not show this as there was no model validation with independent data. LOOCV is not an independent validation, it is a way to determine model parameters that are less subject to outliers.
- The uncertainties on the measurements are never addressed and no effort is made to propagate them through the model in order to provide estimates with associated uncertainties. I believe that at the current stage, this is no longer acceptable. Studies need to reflect what is degree of certainty and quality. Only by doing so, will allow to identify where further improvements need to be made and resources allocated.
- The model of choice was OLS, why? Currently there are several alternatives that allow to model some noise randomness and structure, and consider uncertainties. It is clear from the dispersion of the standardised residuals in figure 7 that the model is incomplete and needs further explaining variables. It also only takes into account objects for which there is a quantity of biomass. What happens over water or over bare ground, would AGB be zero? As important to detect the content it’s also important to detect the lack of content. Also, why these the set of chosen variables?
- Upscaling is central and although the authors touch on this subject in the discussion, it is not clear what is presented in the discussion (figure 9). Is it the spatial aggregation of the predicted AGB into LS size equivalent pixels?
- How TSF was calculated. As it was based on the MCD64A1 C6 product which is a 1km product based on a 4 satellite overpasses. Although it is shown to be a less important variable, the variable measurements could be severely underestimated when considering the frequency and geolocation errors of the satellite observations.
Detail remarks
Line 46 – I assume (Lu 2006) is a reference that was correctly formatted according to the journal specs.
Line 176 to 182 – The modelling needs more explanation as it is not clear from how the results are presented. Was modelling based on a single model (I assume not) or by fuel class
Figure 5 would benefit from adding the number of cases in each scatter plot. Furthermore, this information is nowhere to bee seen in the paper - the reader is not informed on how many objects were identified in each category. In addition, it would benefit from adding the equation line and not the bisection line (in green).
Section 3 – Missing information on the model’s parameters. As the model is a central part of the study, it deserves more analysis and detail on the statistical outputs.
Figure 7 – Not clear if the standardised AGB in the x-axis related to observed or predicted
Reviewer 2 Report
Review of manuscript fire-956443
Title: Instantaneous pre-fire biomass and fuel load measurements from multi-spectral UAV mapping in southern African savannas
This manuscript describes efforts to map fuel load across savanna fuel types for areas in southern Africa. The authors combine field measurements from fuel transects, high resolution multispectral UAV imagery and meteorological data and time since fire data to develop fuel load regression models for several surface fuel bed components. The study is successful in generating maps of fuel load of reasonable accuracy. They then demonstrate the potential for extension of this modeling approach to predict fuel loads over larger areas, leveraging more commonly available LANDSAT imagery.
The manuscript is well written and straightforward to read. The methods are well described, and the topics covered are appropriate for this journal. There are only a few suggestions that I might make for improving it, but as these do not amount to more than minor revisions, I recommend this paper for acceptance, with minor revisions.
My suggestions for improvements are as follows:
Extension in area through leveraging of satellite data – A common issue with efforts to scale up local mapping approaches to larger areas is that the underlying data may be fundamentally different in nature. To strengthen this portion of the discussion, which is important for making this paper more relevant over larger areas, the authors should add more detail regarding the degree to which the near infrared band from the LANDSAT and Sentinel data and NDVI derivatives relate to those in the sensors used in the UAV mapping. Some citations from the literature should be included here as well.
Representativeness and applicability of sites and data used – The authors collected both early and late dry season data, for two sites with different climate regimes, and applied the same regression models, with reasonable results. However, it is not clear to what degree these sites and climatic conditions are applicable, either in space, or within the climatic variability, of savanna landscapes in general within southern Africa. It would strengthen the paper to be more explicit regarding the likely areas where this modeling approach could be reasonably expected to be applied. I would recommend an additional figure which relates these sites to other similar areas in the region, and which illustrates how the measured conditions fit within the climatic context of that broader region.
Reviewer 3 Report
General comments:
In this manuscript, the authors propose a study on the estimation of fuel load from high resolution multispectral observation collected from UAV complemented with the ERA-5 reanalysis dataset. The final output of the work is a model for fuel load estimation that is proposed for application on Sentinel-2 observation.
The subject is of particular importance for the fire science community as it would greatly help the development of satellite based fire emission model. I would however not recommend the publication of this manuscript as it is. I proposed below several points to address.
I did not get how you match the transect data between ground sampling and UAV images. There is a middle 5m section defined at line 164 that I did not see defined earlier and cannot understand. Also, a transect is a path. Why is it coming with 2 dimensions (eg 50x10m)? If 10 is the width used in the data extraction from the maps, why is it double the size of the largest quadra (5m) used for the ground sampling.
Still regarding the data collection, you might have lot of variability along transect both with ground sampling and image-derived fuel class area coverage. Could this be used in the OLS?
I found the extrapolation to sentinel data not enough developped. This is one important point of your conclusion, however this is only brought as an example of application in the discussion section with no proper validation. And again, as for the UAV data, the explanation on the dimension of the training data is confusing (line 260-261).
Finally, I found the manuscript difficult to read in some part. Not being a native english speaker, please consider my rephrasing suggestion as they are (see below in the following specific comments).
Specific comments:
Line 11: AGB is not defined at this point yet.
Line 43: could you better explain the effect of biome dissimilarity on AGB estimation?
Line 44: consider mentioning here the resolution size of the high resolution AGB model.
Line 46: Lu 2006 is not in the same format as other reference.
Line 51: I do not get the problem with scaling up.
Line 83-84: consider rephrasing.
Line 94: as in general comment. I do not understand the definition of your transect.
Line 133: consider mentioning the bands specs.
Line 139: “Each map”. May “Each plot selection” would be better
line 148: “was corrected with”
line157: define the reflectance in the RedEdge
line 159-167: how do you report the transect location on the map? GPS? If so with which accuracy. The is probably here that the middle section should be better defined.
Line 161: the combination of the two methods is not clear.
Line 168: mention ERA-5 resolution.
Figure 4: Could you report transect path on the map together with ground sampling?
Line 178: “somewhat” not necessary here.
Line 194: consider defining fine class (=grass + litter)
Line 196: “which are predominantly consumed in savanna”do you have a reference?
Figure 6: mention origin of the data. (ie Ground sampling)
Line 199: I do not understand the consequence of your data merging? Whay is this coming as the first point of the discussion. Is this really important?
Line 260-264: you need to develop this part much more if you want to make it a possible application of your work. Using input of fuel area coverage derived from satellite observation is probably requiring to redo the OLS regression?
